# Intranasal Therapy in Palliative Care

**DOI:** 10.3390/pharmaceutics16040519

**Published:** 2024-04-09

**Authors:** Anna Ingielewicz, Robert K. Szymczak

**Affiliations:** 1Department of Emergency Medicine, Faculty of Health Science, Medical University of Gdansk, Mariana Smoluchowskiego Street 17, 80-214 Gdansk, Poland; robert.szymczak@gumed.edu.pl; 2Hospice Foundation, Kopernika Street 6, 80-208 Gdansk, Poland

**Keywords:** palliative medicine, intranasal drug delivery, end-of-life care

## Abstract

In recent years, the use of the intranasal route has been actively explored as a possible drug delivery method in the palliative patient population. There are reports demonstrating the effectiveness of nasally administered medications that are routinely used in patients at the end of life. The subject of this study is the intranasal drug administration among palliative patients. The aim is to summarize currently used intranasal therapies among palliative patients, determine the benefits and difficulties, and identify potential areas for future research. A review of available medical literature published between 2013 and 2023 was performed using online scientific databases. The following descriptors were used when searching for articles: “palliative”, “intranasal”, “nasal”, “end-of-life care”, “intranasal drug delivery” and “nasal drug delivery”. Out of 774 articles, 55 directly related to the topic were finally selected and thoroughly analyzed. Based on the bibliographic analysis, it was shown that drugs administered intranasally may be a good, effective, and convenient form of treatment for patients receiving palliative care, in both children and adults. This topic requires further, high-quality clinical research.

## 1. Introduction

There is an increasing interest in non-invasive drug administration around the world [1]. Modern therapies are expected to be safe, non-invasive, and effective. The method of drug administration should be easy to use, well tolerated by the patient, and result in achieving the expected drug concentration in the target tissue. The above promotes satisfactory cooperation with the patient [2,3].

One of the most intensively researched methods of drug delivery is the intranasal route. It enables effective transport of substances directly to the brain and systemic circulation, provided that appropriate chemicals are used [4,5]. In recent years, the area of knowledge regarding the supply of drugs via the transmucosal route, especially intranasal, has been rapidly developing [6,7,8,9]. The intranasal route of drug administration has great potential and promising prospects for further development. Intranasal administration was studied in the treatment of a wide variety of ailments and diseases, such as depression [6], eating disorders [7], obesity [8], hormonal disorders, treatment of primary brain tumors [9], neurodegenerative processes (Parkinson’s disease [7], Alzheimer’s disease [10], Huntington’s disease [11]), coagulation disorders [12,13], treatment of infections (vaccines [14]), migraine [15], and addictions [4]. There are also reports describing the use of systemic treatment by the intranasal route among palliative care patients [16]. There is a growing demand for alternative methods of drug administration for patients at the end of life [2]. One factor behind the increased interest in alternative routes of drug administration in palliative care, especially among community-based settings, was the coronavirus (COVID-19) pandemic [17].

Palliative care patients are those receiving symptomatic treatment for a life-limiting illness [18]. The term “palliative care patient” most often refers to patients with advanced or metastatic cancer. However, it is worth distinguishing a group of non-cancer diseases that lead to a terminal condition, which include the following, among others: end-stage renal failure, severe liver failure, or progressive and incurable neurological conditions [19]. Patients at the end-of-life experience similar problems and distressing symptoms. Dyspnea, pain, and agitation are often observed [20,21]. Depression and anxiety are common. An inability to carry out daily activities, including taking scheduled medication is observed. The care of a terminally ill patient is mainly the responsibility of the family, relatives, and health care professionals [22]. Considering the wide use of intranasal route in different medical conditions, it appears that the group of palliative care patients could benefit significantly from this form of drug administration [2].

The aim of this review is to summarize currently used intranasal therapies among palliative patients. The objective of this publication is also to outline the benefits and difficulties associated with the use of intranasal medications for symptomatic treatment in a group of patients undergoing palliative treatment. Another goal is to identify potential areas for future research on intranasal therapy in terminal care patients. 

## 2. Materials and Methods

The available bibliography on the research topic was reviewed. The study was conducted from December 2023 to February 2024. The following descriptors: “palliative” and “intranasal”, “palliative” and “nasal”, “oncology” and “intranasal”, “end-of-life care” and “intranasal drug delivery” and “nasal drug delivery” were used to search online databases such as MEDLINE Complete, Cochrane Central Register of Controlled Trials, Cochrane Database of Systematic reviews, Cochrane Methodology Register. At this stage, the following criteria were included: papers available in full-text versions, concerning children or adults and written in English. Due to the fact that we wanted to summarize the most up-to-date medical knowledge, an important criterion for selecting articles was the date of publication from the beginning of 2013 to the end of 2023.

The articles obtained during the search were initially selected based on their title and an analysis of the content of the abstract. Duplicate articles were removed. To select the final group, the remaining articles were read in their entirety and the same inclusion criteria were applied as before. This allowed us to exclude publications repeating the same conclusions as well as those that did not apply to the group of patients under study. The search and selection process were carried out by two independent researchers, and any disagreements were resolved by consensus. Out of 774 articles, 55 directly related to the topic were finally selected and thoroughly analyzed. The process of selecting articles from identification in the database to inclusion in the review is presented in Table 1.

Two independent researchers processed and synthesized the data. The included works were critically analyzed, and the results were interpreted and assessed. During this process, data were obtained to answer the research questions included in the objectives of our review. 

## 3. The Nasal Cavity as a Promising Space for Drug Delivery

In the context of the structure and physiological diversity, the nasal cavity is perceived as a promising space for drug supply. Drug absorption through the nose occurs through several routes and mechanisms [2]. The human nasal cavity consists of the following three regions: vestibular, olfactory, and respiratory (Figure 1). 

The vestibular region, which is a small anterior part of the nasal cavity, covers an area of approximately 0.6 cm^2^ and plays a marginal role in the absorption of drugs. 

The olfactory region is located in the upper part of the nasal cavity, occupies an area of approximately 10 cm^2^, and is covered with the olfactory epithelium. The tissues that make up this epithelium allow molecules to be transported directly to the brain (nose-to-brain route). The penetration of molecules directly into the brain tissue through the olfactory epithelium takes place via different forms of transport: intracellular (neuronal endocytosis and exocytosis), extracellular (paracellular and through the blood and lymphatic vessels), and transcellular [2]. The transport of molecules depends on their physicochemical properties, mainly hydro- or lipophilicity, molecular weight, and degree of ionization [7]. Lipophilic substances can pass easily through cell membranes (transcellular transport). Hydrophilic molecules reach the cerebrospinal fluid thanks to a concentration gradient (paracellular transport). An important barrier to paracellular transport is the existence of a tight junction of the epithelial layer. The presence of P-glycoprotein-containing efflux pumps in the olfactory epithelium and endothelial cells surrounding the olfactory bulb hinders the penetration of molecules. The function of the efflux pumps is to limit the expansion of the xenobiotic into the central nervous system (CNS) by excreting it back into the nasal cavity [23]. Despite this phenomenon, most of the drug given intranasally is absorbed into the brain via the neuronal route and a smaller amount via the systemic route [24]. The molecules absorbed via the neuronal route are transported by axonal transport [23]. This process involves the olfactory nerve, which passes through the cribriform plate and reaches the olfactory bulb and other brain area. The involvement of trigeminal nerve fibers is also important in neuronal transport. The V1 and V2 branches of trigeminal nerve innervate the nasal cavity and transmit drug molecules to the brain stem both intracellularly and extracellularly [7]. Intranasal administration of the molecules allows them to penetrate brain tissue via the olfactory and trigeminal pathways bypassing the blood-brain-barrier (BBB) [25,26]. This is particularly important because the BBB is a very tight, selectively permeable membrane that helps maintain homeostasis in the CNS environment and protects the CNS from exposure to xenobiotics and toxic substances [13,14,27]. The BBB has been proven to resist the penetration of approximately 98% of small-molecule drugs and 100% of high-molecular-weight substances [24].

The largest part of the nasal cavity is the respiratory region, occupying an area of approximately 130 cm^2^. It is covered with epithelium, which contains cilia and produces mucus. The respiratory region has various functions, in particular filtering, humidifying, and warming inspired air. The function of respiratory epithelial structures is to contribute to mucociliary clearance [2]. Mucociliary cleansing is responsible for the removal of foreign substances. This process in a healthy nose takes about 12–20 min [4]. The respiratory epithelium is a highly vascularized tissue, which means that molecules can be absorbed directly into the vessels and from there into systemic circulation [14].

## 4. Optimizing Intranasal Drug Delivery

The intranasal administration of intravenous formulations of several drugs (fentanyl, sufentanil, ketamine, hydromorphone, midazolam, haloperidol, naloxone, glucagon) may be an effective alternative to intramuscular or intravenous administration [26]. In a paper by Lam et al. (2020), it was stated that formulations available for intravenous delivery can be used for nasal administration using appropriate devices such as a mucosal atomization device (MAD) [2]. Nevertheless, creating delivery systems that facilitate the supply of molecules to the appropriate tissue is a rapidly developing area of scientific research [1,26,27].

The intranasal systems are still being researched and improved upon to deliver the drug to the brain and achieve the most optimal effect. According to expert opinion, nanocarriers increase part of the dose delivered intranasally. Interesting conclusions are reached in a study by Emad et al. [23] in which some nanocarriers are described. Particle encapsulation was shown to protect drug molecules from precipitation and destabilization by enzymatic agents of the nasal cavity environment. Transferosomes, nanoparticles, microemulsions, nanoemulsions, and liposomes have been proven to be effective in delivering drugs from the nose to the brain. Specially designed polymer-lipid-based nanoparticle for intranasal administration facilitates bypassing the BBB, but also can reach the systemic circulation via vessels of the nasal mucosa, thus avoiding first-pass effect through the gastrointestinal tract and liver [23].

Size, charge, and organicity are important when designing nanoparticles. To illustrate: large (>900 nm in diameter), hydrophobic or highly charged particles will be difficult to diffuse through the mucosa. Pharmaceutical researchers focus on the physicochemical properties of drugs to achieve low molecular weight, high lipophilicity, and good water solubility [4].

In addition to the transport of the drug to the brain, the time rate of drug absorption is also important. There are mucoadhesive agents, ciliostatics or biogels used to prolong the residence time of the drug on the nasal mucosa for better absorption [14,25]. Some reports indicate that microemulsions are effective nanosystems for delivering hydrophobic drugs to the brain. It was observed that the addition of a mucoadhesive agent further increased the concentration of a given drug in brain tissue [28]. There are also studies that attempt to combine different systems, for example, mucoadhesive and mucopenetrating properties on the same particles [29]. Promising, although preclinical, are the results of Di Gioia et al. (2023) study of drug encapsulation in nanocarriers using chitosan and its derivatives [27]. It is believed that prolonged drug exposure to the nasal mucosa can be achieved due to the mucoadhesive properties of these carriers and their ability to temporarily open tight intercellular junctions. Another preclinical study worth highlighting is a report by Han et al. (2023) describing gold nanorods administered to the nasal cavity of laboratory mice [25]. This study, based on analytical methods, found that gold nanowires are quickly absorbed in the brain as early as 10 min after intranasal administration. The gold nanorods were observed to enter the brain via the olfactory bulb and then diffuse to higher brain areas within 1 h of exposure. A study of insulin-containing carbon quantum dots showed that their prolonged drug release on the nasal mucosa could be used to treat Alzheimer’s disease [30]. Neurodegenerative diseases (e.g., Alzheimer’s disease, Parkinson’s disease, Huntington’s disease, glioblastoma) are challenging to treat. The use of nanovesicular-mediated intranasal drug therapy to bypass the effect of first-pass metabolism and deliver the drug directly to the target site, i.e., the brain, is a rapidly developing and promising field of research [24].

Although many therapies can to some extent be achieved without the use of drug-loaded nanocarriers, optimizing intranasal drug delivery using nasal permeability agents, gelling agents, or nanocarrier formulations may be critical in the development of new therapies to meet clinical requirements. This topic was the main area of interest in 19 of the analyzed studies (Table 2).

## 5. Research among Adults and Children Receiving Palliative Care

There were more studies conducted in the adult palliative patient population than in the pediatric population (Table 2). Studies on adult patients were characterized by a higher statistical value, i.e., larger number of research centers involved. Detailed data on the type of paper, the number of analyzed patients, the drug tested, and palliative indications are presented in Table 3 and Table 4.

## 6. Problems with Administering Medications in Palliative Patients, New Hope—Intranasal Route

In the treatment of palliative patients, establishing a venous or subcutaneous access route or administering drugs intramuscularly causes suffering to patients and significantly complicates the administration of drugs by family members who are not medical professionals [2,42]. Chronic solutions such as vascular ports are available only to some patients who have a history of or are undergoing chemotherapy. Their use requires the ability to use a special port needle [65]. Complications such as catheter-related infections or accidental damage that are associated with the presence of intravascular devices in patients with permanently impaired immunity (chemotherapy, cancer itself) are additional factors limiting their widespread and trouble-free use in patients at the end of life [49]. Although drugs administered intravenously quickly reach the desired serum concentrations, this method of treatment is sometimes very difficult.

There are many clinical situations in which obtaining intravascular access is very difficult or undesirable. Apart from palliative patients receiving care at home or in a hospice, these may include the pediatric population requiring quick and effective pain treatment or short sedation [34,69] or even people traveling in difficult weather conditions suffering from symptoms of acute altitude illness [85]. In the above situations, administering the drug intranasally may be the method of choice, avoiding unnecessary suffering, and sometimes even the need for the intervention of professional medical staff while maintaining the same effectiveness of the drug as with intravascular administration [3]. In the context of palliative patients, nursing care is very important and is considered one of the strategic factors in improving patient comfort [63]. However, even simple nursing interventions can increase pain intensity and cause procedural breakthrough pain, the incidence of which reaches up to 12–20% [16]. It is very important that in such situations the pain relief or sedation is simple, easy, and convenient for everyone involved in the process [16,31]. Intranasal drug delivery seems to fulfil these requirements.

Intranasal route is widely used in emergency medicine (out of hospital and in emergency department) [86,87]. Drugs administered intranasally play a particularly important role in situations where their quick systemic action is required, and in cases when another route of administration (e.g., intravenous) is for some reason unavailable or difficult to access [2]. Such situations often occur in emergency medicine, especially among the pediatric population, due to a lack of time and technical possibilities to obtain intravenous access, as well as the invasiveness of obtaining intraosseous access [22,34,56]. An additional barrier can be the uncertainty of drug dosages, and this can affect both the pediatric population and cachectic patients at the end of life. In an emergency, intranasal drug administration is a convenient, safe, and fully acceptable alternative for patients [3,8]. Many of the acute medical problems of palliative patients are similar to the emergency patients. These include unpredictable severe pain, acute dyspnea, confusion or psychomotor agitation [20]. In these cases, in elderly cancer patients, immediate administration of rescue medication is necessary. Considering the emergency medicine experience, transmucosal drug administration (including intranasal) is a promising alternative to the currently preferred subcutaneous and intravenous routes among patients with unpredictable breakthrough pain, breathlessness or seizures. A list of doses proposed by the literature is provided in Table 5.

## 7. Intranasal versus Intravenous Route

Current research focuses on assessing the effectiveness of drugs administered intranasally [82,87]. Several studies compared the intranasal and intravenous route of drug administration [78]. It has been proven that the concentrations of various drugs (e.g., fentanyl, dexmedetomidine, diamorphine, ketamine, dexamethasone) in blood serum after intravenous administration do not differ significantly from those after intranasal administration [56,67]. There were some experiments in patients where serum drug concentrations following intravenous administration are compared with intranasal administration [5,67]. The systemic bioavailability of potent glucocorticoids such as dexamethasone and methylprednisolone administered intranasally was shown to be comparable to intravenous bioavailability [5]. The above suggests that intranasal use of glucocorticosteroids is a non-invasive alternative to intravenous drug administration. In a systematic review aimed at determining the dose of intranasal diamorphine in children treated for breakthrough pain, it was found that the analgesic effect ratio for intravenous morphine compared to intranasal diamorphine was 1:1 [67]. A publication on the use of desmopressin in the prevention and treatment of bleeding of various etiologies showed that desmopressin administered intranasally is as effective as that administered intravenously [12]. The use of intranasal desmopressin was described in the treatment of hemophilia A [13], certain subtypes of von Willebrand’s disease, coagulation disorders in uremia, and in palliative patients with end-stage renal and heart failure. There are also studies indicating that intranasal administration of some drugs does not produce a therapeutic effect. Such an example, according to some studies, may be glucagon in some indications. A study evaluating the efficacy of intranasal glucagon versus placebo in healthy volunteers found that adrenocorticoid, somatotropic, and antidiuretic responses were clinically insignificant [77]; however, there is an approved nasal powder formulation of glucagon available for the treatment of severe hypoglycemia [89].

Some reports compared intranasal and intravenous drug administration in animal models and tissue lines [11,26]. Animal models showed that concentrations in brain tissue may be higher after intranasal administration than after intravenous administration [72]. In the case of drugs acting mainly on the CNS, this is an extremely valuable discovery. If we add the following to the above-presented benefits: potentially lower side effects of the systemic effect of the drug, ease of self-administration by the patient or family, and the non-invasiveness (needle-free), the intranasal route of drug administration appears to be a very important alternative in various treatment options [73]. Some authors claim that the intranasal route has an advantage over intravenous route in delivering a large set of drugs to the brain [28]. However, in the context of comparing drug plasma concentrations after intranasal administration to other routes, there is still a noticeable lack of good quality experimental studies.

## 8. Intranasal versus Oral Route

In the population of palliative patients at the end of life, the convenience of using the drug is particularly important [3]. Taking medications orally is much more convenient for patients than intravenous, intramuscular or subcutaneous administration [16]. However, the ability to bypass the first-pass effect through the gastrointestinal tract and liver gives an advantage to molecules administered intranasally over the oral route [25]. In situations where patients cannot take medications orally due to weakness (e.g., cachexia), impaired consciousness or a disease that causes gastrointestinal failure (e.g., intestinal obstruction caused by a tumor), intranasal administration may be a much better option.

## 9. Intranasal Opioids

Patients receiving hospice care are often treated for both severe pain and psychomotor agitation. This treatment involves the use of opioids and benzodiazepines, which are described as substances that can be administered intranasally with good tolerance and bioavailability [42]. For many years, various intranasally administered fentanyl preparations were widely used in this group of patients to treat breakthrough pain, with very good effectiveness [3,42,48,65].

After analyzing selected articles, it was shown that among patients receiving palliative care, the group of drugs administered intranasally best represented in the bibliography are opioids (e.g., fentanyl, diamorphine) (Table 2). There is abundant evidence of their high effectiveness and feasible application, especially in breakthrough pain [65]. It was observed that morphine administered intravenously, and fentanyl administered intranasally or buccally have similarly high analgesic effectiveness, with good treatment tolerance and improved quality of life in the treatment of procedural pain in cancer patients [55]. Administration of intranasal fentanyl or buccal midazolam was accepted by dying patients and their families [51]. Additionally, several authors noticed that patients treated with intranasal and buccal fentanyl had better physical fitness and were more active [49,52].

The use of intranasal opioids is also being investigated among terminal cardiological patients with end-stage heart failure. Reports on this subject confirm the efficacy and safety, as well as the high tolerability of fentanyl and its derivatives used intranasally [39,40]. However, conclusions from reports on the use of fentanyl for the prevention of acute breathlessness among patients at the end of life from non-malignant causes are inconclusive [41]. Good tolerance of this type of treatment has been proven in both adults and children, and the long-term effects of intranasal administration are already known in the adult population [44].

Interesting reports concern the increasing use of intranasal opioids to treat vaso-occlusive pain in sickle cell disease (SCD). Publications cover both pediatric and adult populations and indicate that rapid administration of intranasal fentanyl can shorten the patient’s Emergency Department (ED) stay and reduce overall opioid consumption [36,46,47]. However, these reports require further thorough studies on larger groups of patients and with more profound methodology [90].

## 10. Sedative, Antidepressant, and Antianxiety Drugs

Regarding sedative, antidepressant or antianxiety drugs, ketamine and dexmedetomidine are the most studied (Table 2). There is evidence in the medical literature of the high effectiveness, efficiency, and ease of intranasal use of these drugs [32,43,46]. Interesting findings are presented in the review by Lemus et al. (2022) where, based on a series of case reports, dexmedetomidine was shown to be highly effective in palliative and hospice care [59]. Subsequent reports indicate the effective and safe effects of dexmedetomidine in refractory situations of irritability, dystonia, and insomnia among palliative children [61,83]. A prospective, randomized, doubled-blind study demonstrated that intranasal dexmedetomidine (2.5 mcg kg^−1^) is superior to intranasal ketamine (5 mg kg^−1^) to provide procedural sedation for radiotherapy in children [58]. A meta-analysis of intranasal delivery of analgesia for moderate to severe pain in children showed that intranasal ketamine has similar analgesic efficacy to intranasal fentanyl but induces deeper sedation [34]. Another study of intranasal ketamine for procedural sedation in children by Rached-d’Astous et al. (2023) showed good effects using a dose of 6 mg kg^−1^ in patients being treated for lacerated wounds [82]. However, limitations in methodology of this study suggest the importance of further research [91]. Although numerous studies confirm the effectiveness and safety of intranasal administration of ketamine to children [82,92], it is still formally administered off-label in many countries. Intranasal ketamine especially in combination with intranasal dexmedetomidine has a high efficacy and safety profile in dental procedural sedation in healthy children [35]. In addition, recent reports have identified intranasal ketamine as effective in the treatment of local nasal and sinus mucosal pain caused by oncological radiation treatment [80].

Less research is devoted to intranasal benzodiazepines (e.g., midazolam, lorazepam), but their high effectiveness in terminating status epilepticus in children is widely known and clinically used [56]. The randomized open-label study comparing intranasal versus intravenous lorazepam (0.1 mg kg^−1^) for control of acute seizures in children has shown clinical seizures remission within 10 min in 83 and 80% of patients, respectively [78]. The authors are aware of an ongoing study comparing the sedative effect of midazolam administered by the subcutaneous and intranasal route, in which the plasma concentration of the drug is being investigated [76].

## 11. Limitations of Intranasal Therapy

A limitation of the use of the intranasal route of drug administration is the limited volume of drug that can be administered. The volume of the drug that can be administered into one nostril in adults and children is 1 mL [16] and 0.3 mL [50], respectively. However, some articles indicate that the volume of substance sprayed or dropped into each nostril should not exceed 0.15 mL [2]. This restriction limits the intranasal use of low concentration drugs. Medications should be administered during a few seconds and divided evenly between nostrils. In addition, the presence of mucociliary clearance eliminates drugs from the nasal space quite quickly—within approximately 12–20 min in healthy nose. Additional limitations to this method of drug delivery include possible disease states of the nasal mucosa (e.g., rhinitis, local peri-infective inflammation, allergic edema) or of the vessels (e.g., diabetes) [2].

## 12. Future Perspectives

Issues related to the limitations of intranasal drug delivery are being actively and intensively investigated [24]. Potential solutions include various particle delivery systems. Among them, nanosystems such as liposomes, polymer nanoparticles (nanocapsules and nanospheres), lipid nanoparticles, artificial exosomes, nanometric emulsions, and nanogels are promising [14,26]. Nanosystems are small particles that facilitate the bioavailability of drugs and increase the residence time of the molecule in the nasal cavity, as well as enable their rapid penetration through the nasal mucosa. These features mean that nanosystems can facilitate the transport of drugs from the nose to the brain and peripheral circulation [27].

The number of patients qualified for palliative treatment of various diseases, especially cancer, increases every year [18,93]. Many of them suffer from disseminated disease, with the tumor process often affecting the central nervous system and, as a result, causing brain edema as the disease expands. In the event of cerebral edema, available treatment for hospice patients includes palliative brain radiotherapy and symptomatic treatment involving the administration of drugs that reduce edema of brain tissue. A commonly used drug that reduces cerebral edema is dexamethasone, a synthetic glucocorticosteroid, a fluorinated derivative of prednisone with a long-lasting and powerful effect, which includes the inhibition of capillary permeability, and thus swelling [66]. There are reports confirming good bioavailability and the achievement of similar serum concentrations of dexamethasone after intranasal administration compared to the intravenous route [5]. Studies in animal models suggest even better penetration into the CNS after intranasal administration compared to other routes (intravenous and oral) [7]. Intranasal administration of dexamethasone was discussed in only three articles, one of which concerns experiments on animal models [7], the second on patients with moderate to severe COVID-19 [17], and the third is a study conducted on healthy volunteers [5]. Based on these articles, it can be assumed that dexamethasone administered intranasally is characterized by high bioavailability, effectiveness of targeting the brain, and a high safety profile. Using dexamethasone among palliative patients should be a subject of future research.

While there is scientific evidence supporting intranasal drug administration and its effects on quality of life [42], the impact of changing the method of drug administration to the nasal route in terms of its cost-effectiveness for health care systems has not yet been thoroughly investigated [94].

The authors suggest that future research should focus on several areas. Firstly, drugs that have potential for intranasal use in palliative medicine should be thoroughly investigated for their plasma and brain tissue concentrations. An additional consideration is the need for intranasal application devices that are practical for use by a person with a disability, e.g., a chronically recumbent, physically incapacitated or confused patient. The intranasal preparation must persist long enough in the nasal mucosa, against ciliary-mucosal clearance, while not irritating it. Finally, a clinical assessment based on the examination of patients receiving end-of-life care, in terms of alleviating symptoms associated with severe condition and impending death, is important.

## 13. Conclusions

According to our review there is a significant interest in the intranasal treatment for patients receiving palliative care. Our main conclusions are as follows: Patients receiving palliative care benefit from the use of drugs administered intranasally. For patients in terminal stages of a disease, the supply of drugs via the intranasal route guarantees quick bioavailability and, consequently, high effectiveness, as well as comfort and safety of use.

So far, scientific studies have focused on opioids and sedatives used intranasally. The benefits of the intranasal use of fentanyl, diamorphine, ketamine or dexmedetomidine in these patients are well documented.

This topic requires further research in terms of both clinical and financial effects on the patient and health care system.

## Figures and Tables

**Figure 1 pharmaceutics-16-00519-f001:**
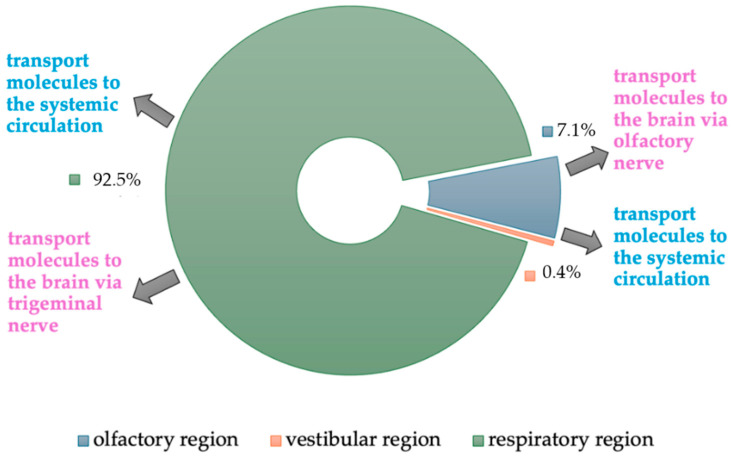
Types of epithelium in the nasal cavity and their role in drug transport.

**Table 1 pharmaceutics-16-00519-t001:** Decision path for selecting articles.

Characteristic	Number of Articles
Articles identified by database search	774
Articles after removing doubles	626
Articles after title analysis	458
Articles after analysis of abstracts	94
Articles after reading the entire text	62
Articles included in the review	55

**Table 2 pharmaceutics-16-00519-t002:** List of publications divided according to their research topics.

Main Research Topic	Number of Papers	Articles
Intranasal opioids	27	[3,22,31,32,33,34,35,36,37,38,39,40,41,42,43,44,45,46,47,48,49,50,51,52,53,54,55]
Intranasal sedatives	10	[6,16,50,56,57,58,59,60,61,62]
Palliative adults	20	[2,6,9,11,16,42,43,44,48,49,51,54,56,57,58,60,63,64,65,66]
Palliative children	10	[3,22,34,53,58,59,61,62,67,68]
Nasal drug delivery systems	19	[1,4,7,10,11,14,23,24,25,26,27,28,69,70,71,72,73,74,75]
Comparison intranasal vs. another route	8	[5,35,47,67,76,77,78,79]

**Table 3 pharmaceutics-16-00519-t003:** List of articles on adult palliative patients divided by number of participants. (Ref.—reference number; No.—number of patients; IV—intravenous; IN—intranasal; PO—oral; SC—subcutaneous; CNS—central nervous system; LVAD—left ventricular assist device).

Adult Palliative Patients					
Ref.	No.	Drug Intranasal	Type of Paper	Palliative Indication	Main Finding
[44]	401	fentanyl	review	breakthrough pain	The most appropriate therapeutic choice is an intranasal opioid (fentanyl spray).
[57]	376	ketamine	review	Depression	Oral or intranasal ketamine may be the most effective for treating depression at home.
[67]	113	diamorphine	systematic review	breakthrough pain	Equianalgesic ratios of IV/PO morphine to IN diamorphine is 1:1 and 1:3, respectively.
[49]	75	fentanyl	original article	breakthrough pain	Long-term use of IN fentanyl is effective and safe (no side effects up to six months).
[17]	60	dexamethasone	study protocol	severe inflammation	Background—therapeutic doses of dexamethasone in the CNS at low IN doses.
[76]	60	midazolam	study protocol	agitation	Hypothesis—equivalent reduction in terminal agitation—IN vs. SC midazolam.
[47]	31	fentanyl	article	pain in sickle cell disease	IN fentanyl and IV morphine similarly reduce pain in vaso-occlusive crisis.
[16]	24	dexmedetomidine	original research article	procedural pain	IN dexmedetomidine—good alternative to SC opioids in procedural pain.
[39]	24	fentanyl	research article	dyspnea	Pretreatment with IN fentanyl may improve dyspnea on exertion in palliative patients with heart failure.
[48]	24	fentanyl	original article	dyspnea	IN fentanyl reduces dyspnea at rest in cancer patients.
[6]	20	ketamine	article	depression	IN ketamine is effective for depression in cancer patients receiving palliative care.
[51]	20	fentanyl	short report	agitation, pain	IN fentanyl—acceptable and well tolerated to control end-of-life symptoms at home.
[41]	19	fentanyl	brief report	dyspnea	No difference between IN fentanyl and placebo in the relief of episodic dyspnea in terminal non-malignant diseases.
[42]	15	fentanyl	original research	procedural pain	IN fentanyl and IV morphine have similarly high analgesic efficacy in procedural pain in cancer patients.
[59]	14	dexmedetomidine	review	sedation, pain	Sedation and analgesia are potential therapeutic applications of IN dexmedetomidine in palliative care.
[37]	3	diamorphine midazolam	case series	pain	IN diamorphine and IN midazolam administered by patients or lay carers at home are acceptable and efficacious.
[40]	1	midazolam sufentanil	case report	pain dyspnea	IN midazolam and sufentanil—effective strategy for palliative care in patients requesting discontinuation of LVAD.
[60]	1	dexmedetomidine	case report	procedural sedation	IN dexmedetomidine—effective in the management of complex wound dressings.
[80]	1	ketamine	case report	mucositis pain in sinonasal carcinoma	IN ketamine—a safe and effective topical treatment for mucositis pain of the sinuses.

**Table 4 pharmaceutics-16-00519-t004:** List of articles on pediatric palliative patients divided by number of participants. (Ref.—reference number; No.—number of patients; IV—intravenous; IN—intranasal; ED—emergency department).

Pediatric Palliative Patients					
Ref.	No.	Drug Intranasal	Type of Paper	Palliative Indication	Main Finding
[50]	23,000	dexmedetomidin, fentanyl, ketamine, midazolam	review	analgosedation	IN analgosedation is a simple, quick, and painless method of treating pain and anxiety in a pediatric emergency department.
[34]	1163	fentanyl, ketorolac, ketamine	systematic review	pain	IN analgesics may be a good alternative to IM and IV analgesics in children with acute moderate to severe pain.
[58]	165	ketamine, dexmedetomidine	original article	sedation	IN dexmedetomidine is superior to IN ketamine to provide procedural sedation for radiotherapy in children.
[35]	128	fentanyl, midazolam, ketamine, dexmedetomidine	research article	procedural dental pain	IN dexmedetomidine-ketamine and IN dexmedetomidine-fentanyl are promising drug combinations with successful anxiolytic and analgesic effects.
[36]	113	fentanyl	research article	pain in sickle cell disease	Adolescents with sickle cell disease, who frequently visit ED due to pain, were more likely to receive IV or IN opioids.
[81]	111	fentanyl midazolam	article	pain in burns	Sedoanalgesia with IN fentanyl-midazolam or IN fentanyl in treatment of childhood burns is safe and highly effective.
[46]	75	fentanyl	research article	pain in sickle cell disease	IN fentanyl is an effective analgesic used to treat episodes of vaso-occlusive pain in children with sickle cell disease.
[82]	30	ketamine	research article	procedural sedation	Single dose of 6 mg/kg of IN ketamine led to effective sedation in 60% of patients.
[53]	16	fentanyl	research article	dyspnea	IN fentanyl may be a safe and effective medication for attacks of respiratory distress in pediatric palliative patients.
[83]	8	dexmedetomidine	case series	dystonia, insomnia	IN dexmedetomidine is a promising approach for sleep disorders or dystonic states in pediatric palliative care children.
[62]	1	dexmedetomidine	case report	irritability	IN dexmedetomidine may be effective in the treatment of refractory irritability.
[61]	1	dexmedetomidine	case report	dystonia	IN dexmedetomidine should be considered for symptomatic treatment of intractable dystonia in children.
[84]	1	dexmedetomidine	case report	insomnia	IN dexmedetomidine may be a safe and effective drug for the treatment of refractory sleep disorders in pediatric palliative patients.

**Table 5 pharmaceutics-16-00519-t005:** Proposed intranasal dose of drugs used in palliative medicine.

Intranasal Drug	Population	Proposed Single Intranasal Dose	Palliative Indications	References
morfine	children	0.1 mg/kg	pain	[67]
	adults	0.1 mg/kg	pain	[67]
diamorphine	children	0.1 mg/kg	pain	[48]
	adults	1.25–2.5 mg	pain	[69]
		2.5 mg	pain	[37]
fentanyl	children	0.5–2 mcg/kg	pain	[34]
		1.5–2 mcg/kg	procedural dental pain, sedation	[35,50]
	adults	100–400 mcg	control symptoms in the dying	[40,51]
		50–100 mcg	pain	[69]
		100 mcg	dyspnea, pain in sickle cell disease	[47,48]
		100–800 mcg	pain	[44]
		50 mcg	dyspnea	[39]
ketamine	children	5 mg/kg	procedural sedation	[58]
		1–4 mg/kg	procedural dental pain	[35]
		6 mg/kg	procedural sedation	[82]
		1.0–1.5 mg/kg	pain	[34]
	adults	50–100 mg	depression	[6]
		50 mg	nasal mucositis pain	[80]
ipratropium	adults	41 mcg	respiratory secretion	[69]
dexmedetomidine	children	2.5 mcg/kg	procedural sedation, dystonia	[58,59,61,83]
		3 mcg/kg	insomnia	[84]
		0.5–4 mcg/kg	sedation	[50]
		1 mcg/kg	procedural dental pain	[35]
	adults	1–1.5 mcg/kg	pain, sedation	[60]
		1.25 mcg/kg	pain, anxiety	[16]
dexamethasone	adults	0.12 mg/kg for 3 days, next 0.06 mg/kg for 7 days	severe inflammation in COVID-19	[17]
midazolam	children	0.2–0.5 mg/kg	sedation	[50]
		0.2 mg/kg	status epilepticus	[56]
	adults	5 mg	agitation	[76]
		2.5–5 mg	agitation	[37]
		0.2 mg/kg	status epilepticus	[88]
lorazepam	children	0.05 mg/kg	status epilepticus	[56]
ketorolac	children	1 mg/kg	pain	[34]

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
