# Peer review of "Intranasal Therapy in Palliative Care"

_pharmaceutics, 2024, doi:10.3390/pharmaceutics16040519_

Round 1

Reviewer 1 Report

Comments and Suggestions for Authors

The review focuses on intranasal delivery in palliative care. It summarises the literature that consists of the descriptors “ Palliative, Intranasal, end of life care and IDD.”. Overall, this is a comprehensive review of a specific niche in IDD, which helps set it apart from the existing publications and reviews on IDD. Here are some minor suggestions/comments.

Some of the sentence structure can be refined to improve readability. E.g.  Page 1, lines 31-34.

Some of the sentences are somewhat controversial. E.g. Page 2, Lines 36. Guarantees rapid transport of substances directly to the brain. Depending on the chemical, the word rapid and guarantee are likely subjective.

The rationale of the paragraph (lines 4 to 53) is unclear. The concepts define in the paragraph is less connected.

The term defined for palliative care in line 55 is controversial. Personally, I don’t think it relates often only to patients with advanced cancer. It is likely applicable to patients at any stage of their disease and is unlikely limited to the end of life.

CSN vS CNS, please make the necessary corrections.

Figure 1, The size of the font is too small. I recommend enlarging them.  

Subsequent writings are fine, and the content is meaningful and informative.

Comments on the Quality of English Language

Other than the Introduction, which I would like to recommend the authors to review and refine, I see no other writing issues thereafter.  

Author Response

Reviewer 1

Comments and Suggestions for Authors

The review focuses on intranasal delivery in palliative care. It summarizes the literature that consists of the descriptors “Palliative, Intranasal, end of life care and IDD.”. Overall, this is a comprehensive review of a specific niche in IDD, which helps set it apart from the existing publications and reviews on IDD. Here are some minor suggestions/comments.

Some of the sentence structure can be refined to improve readability. E.g.  Page 1, lines 31-34.

Answer – The sentence has been corrected to: “Modern therapies are expected to be safe, non-invasive and effective. The method of drug administration should be easy to use, well tolerated by the patient and result in achieving the expected drug concentration in the target tissue. The above promotes satisfactory cooperation with the patient.”

Some of the sentences are somewhat controversial. E.g. Page 2, Lines 36. Guarantees rapid transport of substances directly to the brain. Depending on the chemical, the word rapid and guarantee are likely subjective.

Answer – The sentence has been edited and changed into “One of the most intensively researched methods of drug delivery is the intranasal route.  It enables effective transport of substances directly to the brain and systemic circulation, provided that appropriate chemicals are used.”

The rationale of the paragraph (lines 47 to 53) is unclear. The concepts define in the paragraph is less connected.

Answer - The paragraph has been corrected. The first sentence has been moved to the previous paragraph. The remaining sentences have been corrected. “There is a growing demand for alternative methods of drug administration for patients at the end of life. One of the factors behind the increased interest in alternative routes of drug administration in palliative care, especially in community facilities, was the coronavirus (COVID-19) pandemic.”

The term defined for palliative care in line 55 is controversial. Personally, I don’t think it relates often only to patients with advanced cancer. It is likely applicable to patients at any stage of their disease and is unlikely limited to the end of life.

Answer – We agree with the Reviewer that the term “palliative care patient” refers not only to patients with advanced cancer but also to patients with non-cancer diseases that lead to terminal condition. However, in both above cases, palliative care means that the patient is at the end of life. Such definition of palliative care is presented in the paragraph mentioned by the Reviewer.

CSN vS CNS, please make the necessary corrections.

Answer – It has been corrected.

Figure 1, The size of the font is too small. I recommend enlarging them

Answer – It has been corrected.

Reviewer 2 Report

Comments and Suggestions for Authors

Dr. Ingielewicz and Szymczak has been summarized on the reports and possibility of nasal drug application for the palliative care. This topic can help researchers adapting the new medication for palliative patients using intranasal drug delivery method. Moreover, this review article may lead to expand the versatility of intranasal formulations.

The manuscript is well written and summarized well about the advantages and challenging issue for the intranasal administration.

At present, there are still few investigations reported, and the manuscript suggests that further investigation is needed to expand its nasal application for the palliative care. Reviewer thinks it would be more impactful if there is a section that summarized what specific considerations should be made for improving the practical use of nasal application and to enable clinical application.

Author Response

Reviewer 2

Comments and Suggestions for Authors

Dr. Ingielewicz and Szymczak has been summarized on the reports and possibility of nasal drug application for the palliative care. This topic can help researchers adapting the new medication for palliative patients using intranasal drug delivery method. Moreover, this review article may lead to expand the versatility of intranasal formulations.

The manuscript is well written and summarized well about the advantages and challenging issue for the intranasal administration.

At present, there are still few investigations reported, and the manuscript suggests that further investigation is needed to expand its nasal application for the palliative care. Reviewer thinks it would be more impactful if there is a section that summarized what specific considerations should be made for improving the practical use of nasal application and to enable clinical application.

Answer – Thank you for your favorable review and suggestion to developing the section concerning specific considerations for improving the practical use of nasal application in palliative care. Our suggestions for further developments in the field of intranasal drug administration are given in the new paragraph at the end of section - 12. Future perspectives.

Reviewer 3 Report

Comments and Suggestions for Authors

Manuscript reviews nasal drug delivery in palliative care, summarising the related literature sources published in the last decade. The review article on this topic presents significant contribution in this field and as such is interesting to the readership of this journal. However, there are some major and minor issues that need to be addressed by the aouthors to improve the quality of the manuscript.More particularly:

1)        As for descriptors used when searching for articles, authors should also check „nasal“ and „nasal drug delivery“ to make sure that all relevant articles have been taken into consideration.

2)        Lines 68-70; „The objective of this publication is also to determine the benefits and difficulties associated with the use of intranasal medications for symptomatic treatment in a group of patients undergoing palliative treatment.“ – „determine“ should be replaced with „review“ or „outline“.

3)        Lines 70-71; „Another goal is to identify potential areas for future research on intranasal therapy in terminal care patients“ – this goal should be more thoroughly elaborated in the manuscript.

4)        Lines 107-108; „The blood-brain barrier (BBB) is a very tight, selectively permeable membrane“- authors should review this sentence reassessing the appropriatnes of the term „membrane“ for BBB.

5)        References in the Table two should be listed in increasing order.

6)        Table 2; „Intranasal systems“ should be replaced with nasal drug delivery systems“

7)        Table 3, „List of articles divided according to number of participants“ – the title of the table does not reflect its content. Actually, articles are divided according to population (Adults and children). Moreover, information listed in the table are very limited (i.e. number of patients, drug, type of paper and palliative indication). In reviewer's opinion, Table 3 should be extended with a column reporting the main findings of each of the listed article to make the table appropriately informative for the reader. In addition, table should be checked for the incorrectly spelled names of drugs (e.g. dexmedetomidyna, fenatnyl, alfentanil, ketamina).

8)        Table 4; authors should check the appropriatness of unit „mcg“ in the manuscript. In addition „respiratpry“ should be replaced with „respiratory“; „dexametasone“ should be replaced with „dexamethasone“; „several“ should be replaced with „severe“...; formating of the text should be checked (i.e. space between the number and unit).

9)        Line 173; it is not clear what does „donor administration“ stand for.

10)   Line 174; „systems“ – refers to delivery systems?

11)   Line 180-181; „This is possible due to carrier molecules that facilitate the penetration of the blood-brain barrier.“ Authors should express this statement more precisely to make literature-supported distinction between: 1) direct nose to brain delivery avoiding BBB and 2) potential absorption of drug nanocarriers to systemic circulation thus reaching BBB.

12)   Lines 183-184; „mucoadhesive agents to facilitate transport through the nasal mucosa“ – do mucoadhesive agents really facilitate transport through nasal mucosa? Authors should revise this statement and include mucopenetrative delivery systems to adequatelly cover this issue.

13)   Lines 201-204; „Although many therapies can to some extent be achieved without the use of carrier molecules, optimizing intranasal drug delivery using nasal permeability agents, gelling agents, or nanocarrier formulations may be critical in the development of new therapies to meet clinical requirements. This topic was the main area of interest in 19 of the analyzed studies.“ The term „carrier molecules“ is not appropriately used in this sentence. Authors should revise it to make the comprehansion easier. What is more important, the main results of analysed studies should be reported and discussed in the manuscript.

14)   Lines 227-279; „There are also studies indicating that administering some drugs intranasally does not produce a therapeutic effect. Such an example, according to some studies, may be glucagon.“ – this sentence needs to be revised. In this form it sounds missleading since there is approved nasal powder formulation of glucagon (Baqsimi), available for the treatment of severe hypoglycemia.

15)   Lines 303-304; „Administration of intranasal fentanyl or buccal midazolam was accepted by dying patients and their families.“ – this statement under the title „Intranasal versus oral route“ needs to be revised. Namely, intranasal fentanyl is a substitute for parenteral rather than oral fentanyl administration.

16)   Lines 369-371; „The volume of the drug that can be administered into one nostril in adults and children is 1ml [19] and 0.3 ml [50] respectively.“ Noumerous literature sources report significantly smaller formulation volumes that can be administered per nostril (e.g. Advanced Drug Delivery Reviews 160 (2020) 234-243, cited in this manuscript). Authors should discuss this in the manuscript.

17)   Line 415; „shown“ should be replaced with „have shown“.

Comments on the Quality of English Language

Minor editing of English language is required.

Author Response

Reviewer 3

Comments and Suggestions for Authors

Manuscript reviews nasal drug delivery in palliative care, summarising the related literature sources published in the last decade. The review article on this topic presents significant contribution in this field and as such is interesting to the readership of this journal. However, there are some major and minor issues that need to be addressed by the aouthors to improve the quality of the manuscript.More particularly:

As for descriptors used when searching for articles, authors should also check „nasal“ and „nasal drug delivery“ to make sure that all relevant articles have been taken into consideration.

Answer – thank you for highlighting the additional search descriptor for scientific articles, thanks to which we were able to enrich our work with several interesting publications.

Lines 68-70; „The objective of this publication is also to determine the benefits and difficulties associated with the use of intranasal medications for symptomatic treatment in a group of patients undergoing palliative treatment.“ – „determine“ should be replaced with „review“ or „outline“.

Answer - The text has been corrected according to the reviewer's suggestion.

Lines 70-71; „Another goal is to identify potential areas for future research on intranasal therapy in terminal care patients“ – this goal should be more thoroughly elaborated in the manuscript.

Answer - Our suggestions for further developments in the field of intranasal drug administration are given in the new paragraph – the last paragraph in the section (12. Future perspectives).

 Lines 107-108; „The blood-brain barrier (BBB) is a very tight, selectively permeable membrane“- authors should review this sentence reassessing the appropriatnes of the term „membrane“ for BBB.

Answer – The term "membrane" is very often used in the scientific literature when defining the properties of the BBB. We would like to leave this term in the text.

References in the Table two should be listed in increasing order.

Answer – It has been corrected.

Table 2; „Intranasal systems“ should be replaced with nasal drug delivery systems“

Answer - The text has been corrected according to the reviewer's suggestion.

Table 3, „List of articles divided according to number of participants“ – the title of the table does not reflect its content. Actually, articles are divided according to population (Adults and children). Moreover, information listed in the table are very limited (i.e. number of patients, drug, type of paper and palliative indication). In reviewer's opinion, Table 3 should be extended with a column reporting the main findings of each of the listed article to make the table appropriately informative for the reader. In addition, table should be checked for the incorrectly spelled names of drugs (e.g. dexmedetomidyna, fenatnyl, alfentanil, ketamina).

Answer - The title and icorrectly spelled names have  been corrected according to the reviewer's suggestion. The additional column reporting main findings of the studies has been added. The table has been divided into two tabels, Table 3 - concerning adult patients, Table – concerning pediatric patients

Table 4; authors should check the appropriatness of unit „mcg“ in the manuscript. In addition „respiratpry“ should be replaced with „respiratory“; „dexametasone“ should be replaced with „dexamethasone“; „several“ should be replaced with „severe“...; formating of the text should be checked (i.e. space between the number and unit).

Answer – It has been corrected; It has been checked that the recommended symbol in the United States and United Kingdom for a microgram when communicating medical information is mcg.

Line 173; it is not clear what does „donor administration“ stand for.

                   Answer – It has been corrected. “Donor” has been replaced by “nasal”

Line 174; „systems“ – refers to delivery systems?

                      Answer – It has been corrected into “delivery systems”

Line 180-181; „This is possible due to carrier molecules that facilitate the penetration of the blood-brain barrier.“ Authors should express this statement more precisely to make literature-supported distinction between: 1) direct nose to brain delivery avoiding BBB and 2) potential absorption of drug nanocarriers to systemic circulation thus reaching BBB.

Answer – The whole section 4. “Optimizing intranasal drug delivery” has been revised and corrected including suggestions of the Reviewer.

      Lines 183-184; „mucoadhesive agents to facilitate transport through the nasal mucosa“ – do mucoadhesive agents really facilitate transport through nasal mucosa? Authors should revise this statement and include mucopenetrative delivery systems to adequatelly cover this issue.

Answer – we agree with the Reviewer that this sentence is not sufficiently clear, it has been corrected and expanded in accordance with reviewer’s suggestion –section “4”. Optimizing intranasal drug delivery”.

       Lines 201-204; „Although many therapies can to some extent be achieved without the use of carrier molecules, optimizing intranasal drug delivery using nasal permeability agents, gelling agents, or nanocarrier formulations may be critical in the development of new therapies to meet clinical requirements. This topic was the main area of interest in 19 of the analyzed studies.“ The term „carrier molecules“ is not appropriately used in this sentence. Authors should revise it to make the comprehansion easier. What is more important, the main results of analysed studies should be reported and discussed in the manuscript.

Answer – the term „carrier molecules“ has been replaced by „drug–loaded nanocarriers“. The paragraph has been revised.

      Lines 227-279; „There are also studies indicating that administering some drugs intranasally does not produce a therapeutic effect. Such an example, according to some studies, may be glucagon.“ – this sentence needs to be revised. In this form it sounds missleading since there is approved nasal powder formulation of glucagon (Baqsimi), available for the treatment of severe hypoglycemia.

Answer - the glucagon action in this sentence referred to adrenocorticoid, somatotropic and antidiuretic responses.  In accordance to reviewer's opinion this sentence has been clarified.

Lines 303-304; „Administration of intranasal fentanyl or buccal midazolam was accepted by dying patients and their families.“ – this statement under the title „Intranasal versus oral route“ needs to be revised. Namely, intranasal fentanyl is a substitute for parenteral rather than oral fentanyl administration.

Answer – there was an editing error, it has been corrected. The sentence has been moved to appropriate paragraph.

       Lines 369-371; „The volume of the drug that can be administered into one nostril in adults and children is 1ml [19] and 0.3 ml [50] respectively.“ Noumerous literature sources report significantly smaller formulation volumes that can be administered per nostril (e.g. Advanced Drug Delivery Reviews 160 (2020) 234-243, cited in this manuscript). Authors should discuss this in the manuscript.

Answer – It has been corrected; the reviewer's comment has been taken into consideration in the manuscript

17)   Line 415; „shown“ should be replaced with „have shown“.

Answer – It has been corrected

Reviewer 4 Report

Comments and Suggestions for Authors

Manuscript ID: pharmaceutics-2916350

Type of manuscript: Review

Title: Intranasal Therapy in Palliative Care

Journal: Pharmaceutics

General Comment

The manuscript reports the results of a bibliographic analysis of the literature, from 2013 to 2023, concerning the drugs administered intranasally in palliative care, both for children and adults. Despite the importance of the topic and agreeing with the Authors that nasal administration is decisive for the treatment of many diseases, the review reports the advantages of nasal administration respect to other administration routes without supporting data: “the considerable evidence that shows the effectiveness of nasally administered medications” (lines 13-14) have not been proven. For example, Authors wrote “The intranasal route was proven to be superior to the intravenous route in delivering a large set of drugs to the brain” (lines 283-284): which drugs??? what bioavailability data supports these claims? The reader must have the data available in order to reach the same conclusions as the Authors. A further limit found in this paper is the lack of information about the drug delivery systems used in palliative care: Authors report only of drugs employed, and then affirm the importance of nanoparticles: why? Are conventional formulations ineffective? Furthermore, there are substantial errors related to BBB and nose-to-brain: “The intranasal systems are still being researched and improved upon to deliver the drug to the brain and achieve the most optimal effect [29]. This is possible due to carrier molecules that facilitate the penetration of the blood-brain barrier” (lines 179-181): in the nose-to-brain transport the BBB is not involved!!!

Considering these aspects, the manuscript in the present form does not possess the scientific rigor for its publication in Pharmaceutics. An extensive revision is necessary for removing/reduce the well-know properties and advantages of the nasal drug administration and for highlight the novelty and originality of the manuscript respect to other review on nasal drug delivery.

Other specific comments

Line 101 and 103: please use the word “region“ instead “area.

Line 106: please modify the sentence “It is named the nose-to-brain route because it bypasses the blood-brain barrier” because the name is not related to the bypassing the BBB, but is a direct route that bypasses the blood-brain barrier.

Lines 107-112: Please remove this sentence that could determine a misunderstanding of a relationship between nasal administration and BBB.

Lines 107-112: Please modify this sentence because is the olfactory nerve that pass through the cribriform plate and reach the olfactory bulb and other brain area. The molecules adsorbed are transported by axonal transport.

Section 3: the paracellular transport and reaching the CSF are lacking.

Lines 130-131: the respiratory region has different function and, in particular the humidification & warming of the air. Please modify the generic sentence.

Fig. 1: Please remove the “transport of molecules to systemic circulation in the olfactory region.

The Figure and Tables need to be inserted close to the related text for helping the reader/reviewer.

Table 2 shows a data: please remove.

Table 3 reported number overwritten.

Lines 372-373: Please remove the sentence related to the BBB.

Author Response

Reviewer 4 

General Comment

The manuscript reports the results of a bibliographic analysis of the literature, from 2013 to 2023, concerning the drugs administered intranasally in palliative care, both for children and adults. Despite the importance of the topic and agreeing with the Authors that nasal administration is decisive for the treatment of many diseases, the review reports the advantages of nasal administration respect to other administration routes without supporting data: “the considerable evidence that shows the effectiveness of nasally administered medications” (lines 13-14) have not been proven.

Answer: We agree with the Reviewer that this sentence is misleading, therefore we have changed it into “There are reports demonstrating the effectiveness of nasally administered medications that are routinely used in patients at the end of life.”

For example, Authors wrote “The intranasal route was proven to be superior to the intravenous route in delivering a large set of drugs to the brain” (lines 283-284): which drugs??? what bioavailability data supports these claims? The reader must have the data available in order to reach the same conclusions as the Authors.

Answer: We agree with the reviewer.  We based the quoted sentence on the conclusions of the review (P. C. Pires i A. O. Santos, „Nanosystems in nose-to-brain drug delivery: A review of non-clinical brain targeting studies”, Journal of Controlled Release, t. 270, s. 89–100, sty. 2018, doi: 10.1016/j.jconrel.2017.11.047.) After carefully analyzing the above review again, we decided that the conclusion was too far-reaching. We modified the sentence mentioned by the reviewer so that it would not mislead readers. Currently, the paragraph fragment reads as follows: “Some authors claim that the intranasal route has an advantage over intravenous route in delivering a large set of drugs to the brain [82]. However, in the context of comparing drug plasma concentrations after intranasal administration to other routes, there is still a noticeable lack of good quality experimental studies.”

A further limit found in this paper is the lack of information about the drug delivery systems used in palliative care: Authors report only of drugs employed, and then affirm the importance of nanoparticles: why? Are conventional formulations ineffective?

Answer – We agree with the Reviewer that traditionally available drug solutions used e.g. intravenously are clinically effective in palliative medicine, however, drugs improved with intranasal delivery systems have proven to be more effective.

Furthermore, there are substantial errors related to BBB and nose-to-brain: “The intranasal systems are still being researched and improved upon to deliver the drug to the brain and achieve the most optimal effect [29]. This is possible due to carrier molecules that facilitate the penetration of the blood-brain barrier” (lines 179-181): in the nose-to-brain transport the BBB is not involved!!!

Answer: All parts of the review regarding the BBB and nose-to-brain transport have been revised and improved in line with the valuable suggestions of the Reviewer.

Considering these aspects, the manuscript in the present form does not possess the scientific rigor for its publication in Pharmaceutics. An extensive revision is necessary for removing/reduce the well-know properties and advantages of the nasal drug administration and for highlight the novelty and originality of the manuscript respect to other review on nasal drug delivery.

Answer - The manuscript has been revised according to the suggestions of the Reviewer. Our manuscript refers to a narrow and very specific group of patients, i.e. patients at the end of life. We believe that highlighting the needs of this group of patients by providing a more in-depth knowledge of the best possible routes of intranasal drug administration will provide significant support in the management of their intractable symptoms.

Other specific comments

Line 101 and 103: please use the word “region „instead area.

Answer – It has been corrected.

Line 106: please modify the sentence “It is named the nose-to-brain route because it bypasses the blood-brain barrier” because the name is not related to the bypassing the BBB but is a direct route that bypasses the blood-brain barrier.

Answer – The sentence has been corrected according to the suggestion of the Reviewer. 

Section “3. The nasal cavity as a promising space for drug delivery”

- Lines 107-112: Please remove this sentence that could determine a misunderstanding of a relationship between nasal administration and BBB.

- Lines 112-114: Please modify this sentence because is the olfactory nerve that pass through the cribriform plate and reach the olfactory bulb and other brain area. The molecules adsorbed are transported by axonal transport.

- Section 3: the paracellular transport and reaching the CSF are lacking.

- Lines 130-131: the respiratory region has different function and, in particular the humidification & warming of the air. Please modify the generic sentence.

Answer – The entire section "3. The nasal cavity as a promising space for drug delivery” has been reworked and improved in accordance with the Reviewer's suggestions.

Fig. 1: Please remove the “transport of molecules to systemic circulation in the olfactory region.

Answer – It has been corrected.

The Figure and Tables need to be inserted close to the related text for helping the reader/reviewer.

Answer – It has been corrected.

Table 2 shows a data: please remove.

Answer – It has been corrected.

Table 3 reported number overwritten.

Answer – It has been corrected by dividing the table.

Lines 372-373: Please remove the sentence related to the BBB.

Answer – It has been removed.

Round 2

Reviewer 3 Report

Comments and Suggestions for Authors

Authors have addressed all issues raised by the Reviewer.

The following issue appear in the revised paper:

1) Line 172: "microemulsions are effective nanoparticles" - this statement needs to be revised.

Author Response

Reviewer 3

Authors have addressed all issues raised by the Reviewer.

The following issue appear in the revised paper:

1) Line 172: "microemulsions are effective nanoparticles" - this statement needs to be revised.

Answer: the sentence has been revised and changed into: “Some reports indicate that microemulsions are effective nanosystems for delivering hydrophobic drugs to the brain.”

Reviewer 4 Report

Comments and Suggestions for Authors

The Authors revised carefully the manuscript. Nevertheless, many units are not reported according to IS (minutes vs min (line183), hours vs h (line185)). Some terms are improperly used: nanoparticles (line 172), nanomolecules (line 189). Please change these words with more appropiate terms.

Author Response

Reviewer 4

The Authors revised carefully the manuscript. Nevertheless, many units are not reported according to IS (minutes vs min (line183), hours vs h (line185)).

Answer: we have made the suggested changes

Some terms are improperly used: nanoparticles (line 172),

Answer: the sentence has been revised and changed into: “Some reports indicate that microemulsions are effective nanosystems for delivering hydrophobic drugs to the brain.”

nanomolecules (line 189). Please change these words with more appropiate terms.

Answer: The whole sentence has been revised and changed into: “The use of nanovesicular-mediated intranasal drug therapy to bypass the effect of first-pass metabolism and deliver the drug directly to the target site i.e. the brain, is a rapidly developing and promising field of research.”